# The Recent Advances in Molecular Diagnosis of Soft Tissue Tumors

**DOI:** 10.3390/ijms24065934

**Published:** 2023-03-21

**Authors:** Joon Hyuk Choi, Jae Y. Ro

**Affiliations:** 1Department of Pathology, Yeungnam University College of Medicine, 170 Hyeonchung-ro, Namgu, Daegu 42415, Republic of Korea; 2Department of Pathology and Genomic Medicine, Houston Methodist Hospital, Weill Medical College, Cornell University, Houston, TX 77030, USA

**Keywords:** soft tissue tumor, sarcoma, molecular pathology, translocation, immunohistochemistry

## Abstract

Soft tissue tumors are rare mesenchymal tumors with divergent differentiation. The diagnosis of soft tissue tumors is challenging for pathologists owing to the diversity of tumor types and histological overlap among the tumor entities. Present-day understanding of the molecular pathogenesis of soft tissue tumors has rapidly increased with the development of molecular genetic techniques (e.g., next-generation sequencing). Additionally, immunohistochemical markers that serve as surrogate markers for recurrent translocations in soft tissue tumors have been developed. This review aims to provide an update on recently described molecular findings and relevant novel immunohistochemical markers in selected soft tissue tumors.

## 1. Introduction

Soft tissue tumors comprise a heterogeneous group of tumors with a wide spectrum of differentiation. Soft tissue sarcomas represent less than 1% of all malignant neoplasms [1]. They present diagnostic challenges for pathologists due to the large number of tumor types, the rarity of each tumor type, their considerable morphologic diversity and overlap, and their intrinsic and technological complexity [2]. The classical diagnosis of soft tissue tumors is based on histological findings and ancillary tissue-based tests such as immunohistochemistry. Recent progress in molecular genetics in soft tissue tumors has improved diagnostic precision and refined the classification of these tumors [3].

Molecular genetics and immunohistochemistry are rapidly advancing areas in the diagnosis of soft tissue tumors. Immunohistochemistry plays a crucial role in providing genetic information on tumors. Various types of molecular alterations, including (1) specific chromosomal translocations, (2) specific mutations, (3) gene deletions, (4) gene amplifications, and (5) epigenetic alterations, are efficiently detectable via immunohistochemistry [4]. The classification of soft tissue tumors continues to evolve as new molecular genetic abnormalities are identified [5].

Herein, we review recently described molecular findings and relevant novel immunohistochemical markers in selected soft tissue tumors that can help with diagnosis.

## 2. Etiology

The etiology of most benign and malignant soft tissue tumors is unknown. Minorities are associated with germline mutations in tumor suppressor genes and occur in familial cancer syndromes, such as neurofibromatosis type 1, Gardner syndrome, Li–Fraumeni syndrome, Osler–Weber Rendu syndrome, etc. In rare cases (<10%), genetic and environmental factors, immunodeficiency, irradiation, and viral infections have been linked to the development of malignant soft tissue tumors [1].

Unlike carcinomas, most sarcomas do not arise from well-defined precursor lesions. Events comprising multistage tumorigenesis with progressively accumulated genetic alterations have not yet been clearly identified in most soft tissue tumors [1]. Some sarcomas recapitulate a recognizable mesenchymal lineage (e.g., skeletal muscle). However, they are believed to arise from pluripotent mesenchymal stem cells, which acquire somatic “driver” mutations in oncogenes and tumor suppressor genes [5].

## 3. Classification of Sarcomas Based on Karyotypic Complexity

Genetically, sarcomas can be separated into two major genetic groups [3,6]. One group of sarcomas (20%) is characterized by specific genetic changes and typically simple karyotypes, such as specific chromosomal translocations (e.g., *FUS*::*DDIT3* in myxoid liposarcoma [MLPS]) and oncogenic mutations (e.g., *KIT* mutation in gastrointestinal stromal tumor [GIST]). A majority of sarcomas presenting with unique chromosomal translocations occur predominantly in young patients and tend to have a monomorphic microscopic appearance. Tumors with simple cytogenetic features often have distinctive molecular findings that may be diagnostically useful [7]. Another group of sarcomas (80%) is characterized by non-specific genetic alterations and complex unbalanced karyotypes that are characteristic of severe chromosomal and genetic instability. Sarcomas with more complex cytogenetic features are more common in adults and tend to be microscopically diverse, with a range of cell sizes and shapes within a single tumor [7]. The critical genetic events driving the biology of these sarcomas are largely unknown [8]. Moreover, it is important to recognize that even neoplasms with specific genetic alterations can progress and develop complex karyotypes with tumor progression.

## 4. Molecular Tests

Molecular tests have become increasingly important in diagnosing soft tissue tumors [9,10]. Common molecular methods include conventional cytogenetics, reverse-transcriptase polymerase chain reaction (RT-PCR), and fluorescence in situ hybridization (FISH). Conventional cytogenetics requires fresh tissue and is used to evaluate the entire karyotype. In contrast, FISH and RT-PCR are applied to identify specific translocations/amplifications associated with a given tumor type [11,12]. RT-PCR and FISH are considered complementary; the choice of one over the other is largely dictated by the expertise of the laboratory. Particularly, FISH is highly desirable in the evaluation of round cell sarcomas, spindle cell tumors, well-differentiated adipocytic tumors, and myxoid tumors [11].

More recently, next-generation sequencing (NGS) (massively parallel sequencing or deep sequencing) has emerged as a major tool for identifying known or novel molecular alterations in a wide array of soft tissue tumors [13,14,15,16]. NGS is a highly sensitive method for detecting genetic alterations and can help to diagnose more precisely and characterize more detailed genetic alterations. Additionally, NGS will provide further insight into the pathogenesis of soft tissue tumors and the basis for the development of targeted therapies. Current European Society for Medical Oncology (ESMO) guidelines [17] suggest that the morphologic and immunohistochemical analyses should be complemented by molecular pathology: (1) when the specific histologic diagnosis is uncertain, (2) when the clinicopathologic presentation is unusual, or (3) when the genetic information may have prognostic or predictive relevance. If the molecular analysis is not available, it is recommended that you send it to a reference with all molecular analysis equipment.

## 5. Immunohistochemistry

Over the last decade, molecular genetic findings have led to the development of novel, inexpensive, and quick diagnostic tests with immunohistochemical stains [18]. Recently described immunohistochemical markers are classified into three general categories: (1) protein correlates of molecular genetic alterations (e.g., β-catenin, MDM2, CDK4, H3K27me3, MYC, PDGFRA, RB1, SDHB, SMARCB1 [INI1], and SMARCA4 [BRG1]), (2) protein products of gene fusion (e.g., ALK, BCOR, CCNB3, CAMTA1, DDIT3, FOSB, SS18::SSX, TFE3, and pan-TRK), and (3) diagnostic markers identified by gene expression profiling (e.g., DOG1, ETV4, MUC4, NKX2-2, SATB2, and TLE1).

## 6. Practical Diagnostic Approach to Soft Tissue Tumors

The diagnosis of a soft tissue lesion requires both comprehensive clinical information and adequately processed tissue [9]. The first and most important step toward a correct diagnosis is the careful examination of conventionally stained sections at low magnification. In general, light microscopic assessment of morphology remains the cornerstone of diagnosing soft tissue tumors [19]. Usage of immunostains in the most effective and cost-efficient way requires an algorithmic approach and utilization of the reagents in panels [9]. The selection of a particular molecular test should be based on a specific differential diagnosis and relevant pretest probabilities [7]. Pathologists must exercise caution in interpreting cases because many tumors involve the same gene or even the same translocation [3]. The final diagnosis should be based on a coordinated interpretation of a reasonable morphological impression, clinical and radiologic data, and immunohistochemical and molecular findings that confirm the morphological impression [7].

## 7. Adipocytic Tumors

### 7.1. Spindle Cell Lipoma and Pleomorphic Lipoma

Spindle cell lipoma (SCL) is a benign adipocytic tumor composed of a variable admixture of bland spindle cells, mature adipocytes, and ropy collagen fibers [20]. Pleomorphic lipoma (PL) consists of mature adipocytes with pleomorphic stromal cells and multinucleated floret-like giant cells. SCLs and PLs are considered morphological variations of a single neoplasm. Genetically, SCL and PL are characterized by the deletion of chromosome 13q14, often in combination with the loss of chromosome 16q [21,22,23]. Immunohistochemistry shows that RB1 protein expression is lost in almost all SCL and PL [22]. CD34 is strongly expressed in spindle, pleomorphic, and floret-like giant cells.

Interestingly, loss of 13q14, including *RB1*, is seen in myofibroblastoma and cellular angiofibroma, which are morphologically similar to SCL and PL. The overlapping morphologic and genetic features support the hypothesis that these tumors are related entities, the so-called 13q/*RB1* family of tumors [24,25,26].

### 7.2. Atypical Spindle Cell/Pleomorphic Lipomatous Tumor

In the 2020 WHO classification of tumors of soft tissue and bone, atypical spindle cell/pleomorphic lipomatous tumor (ASPLT) was described for the first time as a single entity. ASPLT is a benign adipocytic neoplasm characterized by mild to moderately atypical spindle cells, pleomorphic cells, mature adipocytes, and lipoblasts within a collagenous or myxoid stroma and ill-defined tumor margins [27]. Genetically, ASPLTs commonly harbor deletions or losses of 13q14, which includes *RB1* and its flanking genes *RCBTB2, DLEU1*, and *ITM2B* [28,29,30,31]. Immunohistochemistry shows that the expression of RB1 in the nucleus is lost in approximately 50–70% of cases [28,29,30].

ASPLTs show no *MDM2* or *CDK4* amplification, which distinguishes them from atypical lipomatous tumor (ALT)/well-differentiated liposarcomas (WDLPS), and dedifferentiated liposarcomas (DDLPS). ASPLTs show molecular differences from SCL and PL, with typically more complex deletions of the 13q region and further deletions/losses of genes flanking *RB1.* However, there is a possibility that these features may represent a disease continuum [29,31,32,33]. ASPLT remains an evolving entity with a need for further understanding.

### 7.3. Atypical Lipomatous Tumor/Well-Differentiated Liposarcoma

Atypical lipomatous tumor/well-differentiated liposarcoma (ALT/WDLPS) is a locally aggressive mesenchymal neoplasm showing adipocytic differentiation with at least focal nuclear atypia in both adipocytes and stromal cells [34]. “ALT” and “WDLPS” are synonyms for explaining morphologically and genetically identical lesions. Genetically, amplification of *MDM2* in 12q15 is almost always present. In addition, several other genes located in the 12q13–q15 region, including *CDK4, TSPAN31, HMGA2, YEATS4, CPM,* and *FRS2,* are commonly coamplified with *MDM2* [35,36]. Immunohistochemically, MDM2 and/or CDK4 nuclear immunopositivity is present in most cases [36].

Immunohistochemistry for MDM2 and CDK4 has now become commonly used. However, these antibodies are not exclusively specific since they demonstrate positive staining in malignant peripheral nerve sheath tumors (MPNSTs) [37] and endometrial stromal sarcomas [38]. Nuclear expression of MDM2 in histiocytes in fat necrosis represents another major pitfall. Moreover, *MDM2* is also amplified in intimal sarcoma, low-grade central osteosarcoma, and parosteal osteosarcoma [39,40,41]. Molecular testing for *MDM2* amplification should be considered in recurrent lipomas, lipomatous tumors with equivocal cytologic atypia, large lipomatous tumors (>15 cm) without cytologic atypia, and lipomatous tumors lacking cytologic atypia in the retroperitoneum, pelvis, and abdomen [42].

### 7.4. Dedifferentiated Liposarcoma

Dedifferentiated liposarcoma (DDLPS) is an ALT/WDLPS that shows an abrupt transition to non-lipogenic sarcoma of variable histological grade, either in the primary disease or in a recurrence [43]. It should be noted that a well-differentiated component may not be identifiable. The dedifferentiated component may show lipoblastic differentiation [44]. Genetically, DDLPS overlaps with ALT/WDLPS and is characterized by the amplification of *MDM2* and *CDK4* [35,45]. As in ALT/WDLPS, several other genes from the 12q13-q21 region and other chromosomal regions are variably coamplified with *MDM2*. The karyotypes and quantitative genomic profiles of DDLPS are often more complicated than those of ALT/WDLPS. Immunohistochemically, nuclear expression of MDM2 and/or CDK4 is observed in the majority of DDLPS cases [46] (Figure 1).

MDM2 immunohistochemistry and *MDM2* amplification can help confirm a diagnosis of DDLPS and distinguish DDLPS from other undifferentiated sarcomas in the relevant clinical context [47]. It has been reported that undifferentiated pleomorphic sarcomas with *MDM2* amplification are in fact DDLPS even in the absence of a well-differentiated LPS component [48].

### 7.5. Myxoid Liposarcoma

Myxoid liposarcoma (MLPS) is a malignant tumor composed of uniform, round to ovoid cells with a variable number of small lipoblasts. These cells are set in a myxoid stroma with a branching capillary vasculature [49]. Genetically, most MLPSs (>90%) are characterized by the t(12;16)(q13;p11) translocation, resulting in the *FUS*::*DDIT3* fusion gene. In approximately 3% of MLPSs, t(12;22)(q13;q12) translocation results in *EWSR1*::*DDIT3* fusion [50]. DDIT3 is a DNA-binding transcription factor involved in adipocytic differentiation [51,52]. The chimeric oncoprotein alters transcription and blocks adipocytic differentiation [53]. More than 50% of MLPS cases carry *TERT* promoter mutations [54], and approximately 25% have mutations that activate the PI3K/mTOR signaling pathway [55]. Recent findings have suggested that immunohistochemistry for DDIT3 is highly sensitive and specific for MLPS, including high-grade (round cell) MLPS cases [56].

The presence of *FUS*::*DDIT3* or *EWSR1*::*DDIT3* fusion helps to distinguish MLPS from other myxoid sarcomas and high-grade MLPS from various round cell sarcomas [57]. *FUS* and *EWSR1* can replace each other and occur in other sarcomas, while *DDIT3* is unique to MLPS. Thus, FISH break-apart probes directed at *DDIT3* serve as a sensitive and specific strategy for MLPS diagnosis [58]. Furthermore, immunohistochemistry for DDIT3 could replace molecular genetic testing in many cases, although limited positivity can be observed in several other tumor types [56].

## 8. Fibroblastic and Myofibroblastic Tumors

### 8.1. Desmoid Fibromatosis

Desmoid fibromatosis is a locally aggressive but non-metastasizing deep-seated (myo)fibroblastic neoplasm with infiltrative growth and a high tendency to local recurrence [59]. The majority (90–95%) of sporadic desmoid tumors result from point mutations of the *CTNNB1* gene on 3p21, which encodes β-catenin [60,61]. A minority of desmoid tumors occur in Gardner syndrome and harbor germline mutations of the *APC* gene on 5q21–q22 [62,63,64]. The activating mutations in *CTNNB1* or inactivating mutations in *APC* interfere with the proteasomal degradation of β-catenin, resulting in the accumulation of β-catenin in the nucleus [65]. Immunohistochemically, nuclear expression of β-catenin is present in ~80% of tumors [66].

Importantly, although aberrant nuclear β-catenin is a helpful finding to support the diagnosis, nuclear β-catenin expression is also seen in superficial fibromatoses and some sarcomas and is consequently neither specific nor fully sensitive for desmoid fibromatosis [67,68]. *CTNNB1* mutation analysis may be useful in small biopsy specimens and/or in cases of equivocal immunostaining for β-catenin [69,70]. Moreover, desmoid fibromatosis with specific mutations in exon 3 of *CTNNB1*, particularly S45F, has a greater tendency to local recurrence [71].

### 8.2. Solitary Fibrous Tumor

Solitary fibrous tumor (SFT) is a fibroblastic tumor with prominent, branched, thin-walled, dilated (staghorn) vasculature [72]. The genetic hallmark of SFT is a paracentric inversion involving chromosome 12q, resulting in the fusion of the *NAB2* and *STAT6* genes [73,74]. Overexpression of the *NAB2*::*STAT6* gene fusion has been reported to induce proliferation in cultured cells and activate the expression of EGR-responsive genes [75]. These results establish *NAB2*::*STAT6* as the defining driver mutation of SFT. Overexpression of *ALDH1A1 (ALDH1), EGFR, JAK2*, histone deacetylases, and retinoic acid receptors may also contribute to tumorigenesis [76,77]. In addition, *TERT* promoter mutations [78] and deletions or mutations of *TP53* [79] are associated with aggressive behavior and dedifferentiation. Immunohistochemically, the *NAB2*::*STAT6* fusion leads to the nuclear expression of STAT6 [80] (Figure 2). Thus, STAT6 immunohistochemistry is a sensitive and specific surrogate for all fusions [72]. CC34 is typically positive.

*IGF2* overexpression is consistently detected in SFTs, regardless of anatomical location, and may be associated with triggering hypoglycemia in some patients [76]. *STAT6* gene is located on the long arm of chromosome 12 in the same region as *MDM2* and is, therefore, coamplified and overexpressed in a subset of DDLPS [81]. Dedifferentiated SFTs show an abrupt transformation into an indistinguishable pleomorphic appearance. Nuclear expression of STAT6 may be lost in dedifferentiated SFT [82]. The behavior of SFTs has been challenging to predict. A new risk stratification model based on patient age, tumor size, necrosis, and mitotic activity has been shown to more accurately predict the prognosis of SFTs [83].

### 8.3. Inflammatory Myofibroblastic Tumor

Inflammatory myofibroblastic tumor (IMT) is a characteristic, rarely metastasizing neoplasm composed of myofibroblastic and fibroblastic spindle cells accompanied by an inflammatory infiltrate of plasma cells, lymphocytes, and/or eosinophils [84]. Genetically, IMTs are heterogeneous. In 50–60% of cases of IMT in children and young adults, the tumors fuse the *ALK* gene on 2p23 with various partner genes, including *TPM3, TPM4, CLTC,* and others [85,86]. *ALK* rearrangement is uncommon in IMTs diagnosed in older adults. *ROS1* and *NTRK3* gene rearrangements are each found in 5–10% of IMTs [87,88,89]. Very rare cases have *RET* or *PDGFRB* gene rearrangements [90]. Epithelioid inflammatory myofibroblastic sarcoma (EIMS) is a rare, aggressive variant dominated by epithelioid cells with amphophilic cytoplasm. EIMS has often been associated with *RANBP2*::*ALK* or *RRBP1*::*ALK* gene rearrangements [90,91]. Immunohistochemically, ALK expression is detectable in 50–60% of IMT cases (Figure 3). The ALK immunostaining pattern varies depending on the *ALK* fusion partner; for example, *RANBP2*::*ALK is* correlated with a nuclear membranous pattern, *RRBP1*::*ALK* with a perinuclear accentuated cytoplasmic pattern, and *CLTC*::*ALK* with a granular cytoplasmic pattern. This is while several other ALK fusion variants illustrate a diffuse cytoplasmic pattern (most commonly seen in IMT) [90].

In ALK-negative cases, IHC for ROS1 and/or molecular assays for non-ALK gene fusions (e.g., *NTRK3*) may be useful [87,90]. *ROS1*-rearranged IMT typically shows a cytoplasmic expression of ROS1 [87]. Highly sensitive ALK antibody clones (5A4 and D5F3) can enhance the detection of the ALK protein in IMT [87]. A persistent partial response to the ALK inhibitor crizotinib has been reported in a patient with ALK-translocated IMT [92].

### 8.4. Low-Grade Fibromyxoid Sarcoma

Low-grade fibromyxoid sarcoma (LGFMS) is a malignant fibroblastic neoplasm with bland spindle cells that grow in whorling growth pattern, a matrix that can be either collagenous or myxoid, and arcades of small blood vessels [93]. Genetically, LGFMSs exhibit a characteristic t(7;16)(q33;p11) translocation that results in the *FUS*::*CREB3L2* fusion oncogene in > 90% of cases [94,95,96,97]. The FUS::CREB3L2 chimeric protein acts as an aberrant transcription factor, causing deregulated expression of *CREB3L2* target genes [97]. Rare cases of LGFMS show the presence of *FUS*::*CREB3L1* or *EWSR1*::*CREB3L1* fusion genes [94,98]. The *MUC4* gene on the long arm of chromosome 3 (3q29) is upregulated in LGFMS [97]. Immunohistochemically, tumor cells show strong, diffuse cytoplasmic expression of MUC4 in 100% of LGFMS cases [99,100] (Figure 4). EMA is also expressed in 80% of cases.

MUC4 is a high-molecular-weight transmembrane glycoprotein expressed on the cell membrane of many epithelial cells. It is a highly sensitive and specific marker for LGFMS and can help distinguish LGFMS from histologic mimics, such as soft tissue perineurioma [93]. If MUC4 is negative or unavailable to confirm the diagnosis, identification of *FUS*::*CREB3L2* (or other uncommon variants) offers molecular genetic support for the diagnosis of LGFMS [93].

### 8.5. Sclerosing Epithelioid Fibrosarcoma

Sclerosing epithelioid fibrosarcoma (SEF) is a rare malignant fibroblastic neoplasm with epithelioid fibroblastic tumor cells arranged in cords and nests with dense sclerotic hyaline stroma. A subset of SEFs is morphologically and molecularly related to LGFMS [101]. Genetically, most cases of pure SEF harbor the *EWSR1*::*CREB3L1* gene fusion [102,103,104]. In rare cases, *EWSR1* is exchanged for *FUS* or *PAX5,* and/or *CREB3L1* for *CREB3L2, CREB3L3*, or *CREM* [105,106]. SEF and LGFMS also have overlapping gene expression profiles. For instance, both tumors demonstrate high expression of MUC4 and CD24 [98]. Immunohistochemically, MUC4 is expressed in 80–90% of SEF cases with strong, diffuse, and cytoplasmic staining [107].

Cases showing hybrid features of SEF and LGFMS are now well described. Most cases of hybrid SEF/LGFMS show *FUS*::*CREB3L2* gene fusion and MUC4 immunopositivity [96]. A small subset of SEFs lacks MUC4 expression. Recently, recurrent *YAP1* and *KMT2A* gene rearrangements have been identified in MUC4-negative SEFs [108]. SEF has been shown to exhibit more aggressive behavior than LGFMS [109].

### 8.6. Infantile Fibrosarcoma

Conventional fibrosarcoma falls into two main categories: adult and infantile types. Infantile fibrosarcoma (IFS) is a malignant fibroblastic tumor that occurs most frequently in infancy [110]. It is a locally aggressive and rapidly growing tumor that rarely metastasizes. Most of these tumors harbor the *ETV6*::*NTRK3* fusion resulting from t(12;15)(p13;q25). NTRK3 is a receptor tyrosine kinase. The *ETV6*::*NTRK3* fusion gene encodes a constitutively activated chimeric tyrosine kinase that signals via the RAS-MAPK and PI3K signaling pathways to drive cellular transformation and oncogenesis [111]. In addition, *EML4*::*NTRK*3 fusion has been reported in rare cases [112]. Alternative gene fusions have been reported in a subset of cases, including *NTRK1, NTRK2, BRAF*, and *MET* [113,114]. Immunohistochemically, IFSs with *NTRK* gene rearrangements are often positive for a pan-TRK antibody [115].

The *ETV6*::*NTRK3* fusion is found in a variety of neoplasms, such as cellular congenital mesoblastic nephroma [111], secretory carcinomas of the breast [116], and salivary gland-type secretory carcinoma [117]. It has been demonstrated that pan-TRK antibodies are not entirely specific for tumors with *NTRK* rearrangements [115]. Hence, FISH is most often used to confirm the diagnosis. Adult fibrosarcoma is an uncommon sarcoma composed of relatively monomorphic fibroblastic tumor cells with variable collagen production and often herringbone architecture. Its diagnosis is based on the principle of exclusion; true examples are exceedingly rare [118].

## 9. Vascular Tumors

### 9.1. Epithelioid Hemangioma

Epithelioid hemangioma is a benign vascular neoplasm consisting of well-formed blood vessels lined by plump, epithelioid (histiocytoid) endothelial cells with abundant eosinophilic cytoplasm and a variable infiltrate of eosinophils [119]. Genetically, epithelioid hemangiomas are characterized by recurrent fusion genes affecting the *FOS* or *FOSB* gene in approximately 50% of the cases. The gene partners for *FOS* include *LMNA, MBNL1, VIM*, and lincRNA [120,121], whereas *FOSB* is often fused to *ZFP36, WWTR1,* or *ACTB* [122,123]. The key event in the pathogenesis is the dysregulation of the FOS family (FOS, FOSB, FOSL1, and FOSL2) of transcription factors through chromosomal translocation. Immunohistochemically, a subset of cases shows FOS or FOSB expression, which may be diagnostically useful [121,123].

Interestingly, angiolymphoid hyperplasia with an eosinophilia subtype (cutaneous epithelioid hemangioma) may be non-neoplastic as they lack *FOS* or *FOSB* gene rearrangement [120]. Although *FOSB* fusions have not been found in cases of angiolymphoid hyperplasia with eosinophilia, the endothelial cells in this tumor are often positive for FOSB [120,124]. *FOSB* fusion-associated tumors, particularly *ZFP36*::*FOSB*, are more commonly cellular and solid, with some cytologic atypia and occasional necrosis [123].

### 9.2. Pseudomyogenic Hemangioendothelioma

Pseudomyogenic hemangioendothelioma (PHE) is a rarely metastasizing endothelial neoplasm commonly occurring in young adult males [125]. It often presents as multiple discontinuous nodules at different tissue levels and histologically mimics a myoid tumor or epithelioid sarcoma (EPS). PHEs have t(7;19)(q22;q13), resulting in *SERPINE1*::*FOSB* gene fusion [126,127]. An alternative *ACTB*::*FOSB* gene fusion was identified in half of the cases [122]. These fusions lead to the upregulation of FOSB, a member of the FOS family of transcription factors that encode leucine zipper proteins that interact with the JUN family to regulate cell proliferation, differentiation, angiogenesis, and survival. Immunohistochemically, nuclear staining for FOSB is present in almost all cases of PHE [124,128]. PHE also shows a nuclear expression of the endothelial transcription factors FLI1 and ERG [129,130]. Approximately 50% of cases are positive for CD31. In addition, SMA is predominantly expressed in one-third of tumors.

The clinicopathologic features and behavior do not differ between PHEs with *SERPINE1*::*FOSB* and *ACTB*::*FOSB* fusions are rare, although tumors with the ACTB variant occur more frequently as solitary lesions [122]. Recently, *FOSB* fusions with *WWTR1* or *CLTC* have been described, the latter in a bone lesion in an adolescent [131,132].

### 9.3. Epithelioid Hemangioendothelioma

Epithelioid hemangioendothelioma (EHE) is a malignant vascular neoplasm composed of epithelioid endothelial cells within a characteristic myxohyaline stroma [133]. Genetically, EHE is characterized by a t(1;3)(p36;q25), resulting in a *WWTR1*::*CAMTA1* fusion in 90% of cases [134,135,136]. Fusion of *WWTR1* with *CAMTA1* leads to dysregulation of the Hippo signaling pathway, such that *WWTR1*::*CAMTA1* resides constitutively in the cell nucleus and drives oncogenic transformation [137]. The *YAP1*::*TFE3* fusion due to a t(X;11)(p11;q13) translocation is observed in 5% of EHEs with a distinctly vasoformative morphology [138]. TFE3, an oncogenic transcription factor involved in other soft tissue tumor translocations, is upregulated as a consequence of the *YAP1*::*TFE3* fusion. Immunohistochemically, EHE with *WWTR1*::*CAMTA1* typically shows a diffusely strong nuclear expression of CAMTA1 [139] (Figure 5). Tumors with *YAP1*::*TFE3* fusion show diffuse nuclear expression of TFE3 [138].

CAMTA1 is a highly sensitive and specific diagnostic marker and can help distinguish EHE with *WWTR1*::*CAMTA1* fusion from cellular epithelioid hemangiomas and epithelioid angiosarcomas. TFE3-positive EHEs occasionally exhibit *WWTR1*::*CAMTA1* gene fusions [140]. These findings show that two chromosomal alterations are not mutually exclusive but rather composable in EHEs.

## 10. Skeletal Muscle Tumors

### 10.1. Alveolar Rhabdomyosarcoma

Alveolar rhabdomyosarcoma (ARMS) is a malignant neoplasm composed of primitive monomorphic round cells with skeletal muscle differentiation [141]. Approximately 85% of ARMSs contain characteristic fusion genes. In fusion-positive ARMSs, *PAX3*::*FOXO1* and *PAX7*::*FOXO1* fusion genes are detected in 70–90% and 10–30% of the fusions, respectively [142,143]. PAX3 and PAX7 represent transcription factors that play an essential role in myogenesis [144]. The *PAX*::*FOXO1* fusion proteins function as oncoproteins, affecting growth, survival, differentiation, and other signaling pathways by activating numerous downstream target genes, such as *MET, ALK, FGFR4, MYCN, IGF1R*, and *MYOD1* [145,146,147]. Immunohistochemically, the nuclear expression of myogenin is strong and diffuse, unlike in embryonal rhabdomyosarcoma (RMS) and other RMS subtypes in which the staining pattern is focal [148,149]. MyoD1 is predominantly expressed in ARMS.

Approximately 20% of ARMS cases are fusion-negative [150]. Fusion-negative ARMS is genetically heterogeneous and may have alternative fusions with other genes (*NCOA1* and *INO80D*). Histologic appearances of ARMS do not predict the presence or type of gene fusion. However, solid growth or mixed embryonal/alveolar patterns show a higher incidence of fusion negativity [150]. Fusion-positive ARMS patients have a poorer outcome than fusion-negative ARMS patients [151].

### 10.2. Spindle Cell/Sclerosing Rhabdomyosarcoma

Spindle cell/sclerosing rhabdomyosarcoma (RMS) is a type of RMS with fascicular spindle cells and/or primitive cells in a prominent hyaline collagenous stroma [152]. Spindle cell/sclerosing RMSs are categorized into three groups based on their genetic background. The first group, congenital/infantile spindle cell RMS, shows gene fusions involving *VGLL2, SRF, TEAD1, NCOA2*, and *CITED2* [153,154]. The second group includes most of the spindle cell/sclerosing RMS in adolescents and young adults and a subset of tumors in older adults, showing the presence of the *MYOD1* mutation [154,155]. The third group shows no recurrent, identifiable genetic alterations. Immunohistochemically, spindle cell/sclerosing RMS typically shows diffuse nuclear staining for MyoD1. Myogenin shows only limited expression in most spindle cells/sclerosing RMSs.

The recently described intraosseous spindle cell RMS shows two gene fusions, *EWSR1/FUS*::*TFCP2* and *MEIS1*::*NCOA2* [155,156]. Congenital/infantile spindle cell/sclerosing RMSs with gene fusions show a favorable clinical progression [157]. *MYOD1*-mutant spindle cells/sclerosing RMSs follow an aggressive clinical course despite multimodality therapy [158].

## 11. Gastrointestinal Stromal Tumor

A gastrointestinal stromal tumor (GIST) is a mesenchymal neoplasm characterized by differentiation to the interstitial cells of Cajal with variable behavior [159]. Approximately 75% of GISTs harbor activating mutations of *KIT*, most commonly in exon 11 (66% overall) or exon 9 (6%); mutations in exons 13 and 17 are rare (~1% each) [160,161]. Approximately 10% of GISTs harbor *PDGFRA*-activating mutations (most frequently in the stomach) [162,163]. The *KIT* or *PDGFRA* oncogene is located on chromosome 4 (4q12) and encodes type III receptor tyrosine kinases [162]. Downstream oncogenic signaling involves the RAS/MAPK and PI3K/AKT/mTOR signaling pathways [161,164]. Many *KIT/PDGFRA* wild-type GISTs have alterations in SDH subunit genes (5–10% overall) [165,166]. Almost all pediatric GISTs are SDH-deficient [165,167]. *SDHA* is the most commonly mutated subunit gene (~35% of SDH-deficient GISTs), followed by *SDHB, SDHC*, and *SDHD* [168]. Immunophenotypically, most GISTs show strong and diffuse expression of KIT (CD117). DOG1 (ANO1) is highly sensitive and specific for GIST and is useful in diagnosing KIT-negative GISTs [169]. SDH-deficient GISTs show a loss of SDHB protein expression, regardless of which SDH gene is mutated [168]. SDHA loss is specific to SDHA-mutant tumors [168].

Patients with *PDGFRA*-mutant tumors have a lower risk of metastasis than patients with *KIT*-mutant tumors [170]. GISTs harboring the *PDGFRA* D842V mutation have been shown to respond to avapritinib—a novel KIT and PDGFRA inhibitor [171]. Recently, the use of SDHB by immunohistochemistry has been used to stratify GIST into an SDHB-retained and an SDHB-deficient group, regardless of whether the responsible mutation was acquired or inherited. This widely available screening approach can facilitate decisions about further molecular testing strategies [172].

## 12. Peripheral Nerve Sheath Tumors

### 12.1. Malignant Peripheral Nerve Sheath Tumor

A malignant peripheral nerve sheath tumor (MPNST) is a malignant spindle cell tumor that shows evidence of nerve sheath differentiation. It often arises from a peripheral nerve, a pre-existing benign nerve sheath tumor, or in a patient with neurofibromatosis type 1 [173]. Genetically, conventional MPNSTs have complex karyotypes. Irrespective of whether they are NF1-associated, sporadic, or radiotherapy-associated, the majority of MPNSTs demonstrate highly recurrent and specific inactivating mutations of polycomb repressive complex 2 (PRC2) components (*EED* or *SUZ12*), *NF1,* and *CDKN2A/CDKN2B* [174,175,176]. Inactivation of the PRC2 results in the loss of histone H3K27 trimethylation (H3K27me3) [177,178]. Immunohistochemically, loss of H3K27me3 expression is more common in high-grade tumors (70–80%) than in low-grade tumors (20–30%) [178,179] (Figure 6). Thus, H3K27me3 can serve as a useful diagnostic marker for MPNST.

At the molecular level, epithelioid MPNSTs differ from conventional MPNSTs. Approximately 75% of epithelioid MPNST cases show *SMARCB1* gene inactivation, resulting in SMARCB1 loss by immunohistochemistry [180]. In contrast to conventional MPNSTs, nuclear staining for H3K27me3 is retained in epithelioid MPNSTs (i.e., normal). MPNSTs showing complete heterologous rhabdomyoblastic differentiation mimic spindle cell RMS. Immunohistochemistry for H3K27me3 reliably distinguishes MPNST with complete heterologous rhabdomyoblastic differentiation from spindle cell RMS [181].

### 12.2. Malignant Melanotic Nerve Sheath Tumor

A malignant melanotic nerve sheath tumor (MMNST) is a rare peripheral nerve sheath tumor composed of tumor cells with features of both Schwann cell and melanocytic differentiation [182]. It is most commonly associated with spinal or autonomic nerves near the midline. MMNSTs are frequently associated with the Carney complex. The majority of MMNSTs have inactivating mutations of the *PRKAR1A* gene on 17q24.2 [183]. PRKAR1A plays a central role in the development of MMNST. Immunohistochemically, PRKAR1A expression is typically lost in MMNST [183,184]. MMNSTs strongly express S100 protein, SOX10, and various melanocytic markers, including HMB45, Melan-A, and tyrosinase.

Psammoma bodies are found in approximately 50% of cases [182]. There are no clinical distinctions between psammomatous and non-psammomatous MMNSTs [183,184]. In addition, their histologic features do not correlate well with their clinical behavior. MMNSTs often show aggressive behavior [185,186]. It is critical to distinguish MMNST from malignant melanoma. The paravertebral location, heavy melanin pigmentation, psammoma bodies, and loss of PRKAR1A expression suggest the diagnosis of MMNST [184].

## 13. Tumors of Uncertain Differentiation

### 13.1. Synovial Sarcoma

Synovial sarcoma (SS) is a monomorphic spindle cell mesenchymal neoplasm with variable epithelial differentiation [187]. SS harbors a unique t(X;18)(p11.2;q11.2) translocation [188], by which one of the SSX genes (*SSX1*, *SSX2*, or *SSX4*) on the X chromosome fuses to *SS18* on chromosome 18 [189]. Approximately two-thirds of SS cases harbor an *SS18*::*SSX1* fusion, one-third harbor an *SS18::SSX2* fusion*,* and uncommon cases harbor an *SS18*::*SSX4* fusion [189,190]. *SS18*::*SSX* functions as an oncogene, and its expression is necessary to maintain the transformed phenotype of SS cells [191,192]. Immunohistochemically, a novel SS18-SSX fusion-specific antibody is highly sensitive (95%) and specific (100%) for SS [193] (Figure 7). Thus, SS18-SSX immunohistochemistry is a useful tool to confirm SS diagnosis, and it can replace molecular testing in most cases [194]. Moderate or strong nuclear staining for the transcriptional corepressor TLE1 is present in the majority of SS cases [195]. However, TLE1 staining is not specific to SS because it can also exist in histological mimics of SS, particularly MPNST and SFT [195].

Interestingly, SS shows correlations between fusion type, tumor histology, and patient sex. Almost all *SS18*::*SSX2* fusion cases show monophasic histology. In contrast, *SS18::SSX1* fusion cases show an approximate 2:1 ratio of monophasic to biphasic SS [191]. Males show a 3:2 ratio of *SS18::SSX1* to *SS18*::*SSX2*, whereas in females, the ratio is close to 1:1. Molecular confirmation (if available) of an *SS18*::*SSX1/2/4* fusion should ideally be performed for optimal diagnostic accuracy [57]. Recently, novel and rare *SSX1* fusions to non-*SS18* genes have been reported in SS [196].

### 13.2. Epithelioid Sarcoma

Epithelioid sarcoma (EPS) is a malignant mesenchymal neoplasm exhibiting partial or complete epithelioid cytomorphology and evidence of epithelial differentiation [197]. Two clinicopathological subtypes are recognized in EPSs: (1) the classic (or distal) form, characterized by its propensity for acral sites and pseudogranulomatous growth pattern, and (2) the proximal-type (large cell) subtype, occurring mainly in proximal/truncal regions and consisting of nests and sheets of large epithelioid cells. Genetically, approximately 90% of both classic and proximal-type patients have *SMARCB1 (INI1)* deletion. The *SMARCB1* gene (also called *BAF47, INI1*, or *SNF5*) at chromosome 22q11.2 encodes a protein part of the SWI/SNF chromatin-remodeling complex present in normal cells. Loss of expression of SWI/SNF chromatin-remodeling complex proteins plays an essential role in tumorigenesis [198]. Immunohistochemically, loss of SMARCB1 expression occurs in the majority of EPS cases [199,200,201]. ERG expression is commonly observed in EPSs, which can lead to confusion with endothelial tumors [202,203,204]. Most EPS cases are positive for cytokeratins and EMA. CD34 is expressed in 50–60% of EPS cases, which helps distinguish EPS from carcinomas.

An extreme minority of EPSs retain SMARCB1 (INI1) protein expression. The biological behavior of SMARCB1 (INI1)-preserved EPS is more aggressive than that of EPS with complete loss of SMARCB1 expression [198].

### 13.3. Extrarenal Rhabdoid Tumor

An extrarenal rhabdoid tumor (ERT) is a highly malignant soft tissue neoplasm composed of characteristic rounded or polygonal rhabdoid cells with glassy eosinophilic cytoplasm containing hyaline-like inclusion bodies, eccentric nuclei, and macronucleoli [205]. It mainly affects infants and children. Morphologically and genetically identical tumors also occur in the kidney and brain. Most ERTs are characterized by biallelic alterations of the *SMARCB1* gene, resulting in a loss of expression of SMARCB1 (INI1). Immunohistochemically, the tumors show loss of SMARCB1 (INI1) expression [206,207]. In addition, SALL4 and glypican-3 expressions are frequently observed in ERTs [208,209].

A loss of SMARCB1 expression is also present in EPS, epithelioid MPNST, and myoepithelial carcinoma. When a tumor histologically similar to ERT occurs in adults, pathologists should first consider the possibility of malignant melanoma or other tumor types. Familial cases are typically associated with germline mutations in the *SMARCB1* gene [210,211]. Mutations and/or loss of the *SMARCA4* gene in 19p13.2 have been reported in rare rhabdoid tumors with retention of SMARCB1 expression [212].

### 13.4. Alveolar Soft Part Sarcoma

Alveolar soft part sarcoma (ASPS) is a rare tumor of uncertain histogenesis predominantly affecting the deep soft tissues of the extremities [213]. ASPS features variably discohesive epithelioid cells arranged in nests, resulting in a distinct alveolar growth pattern. It is characterized by a specific translocation, der(17)t(X;17)(p11.2;q25), resulting in *ASPSCR1*::*TFE3* gene fusion. The *ASPSCR1*::*TFE3* fusion protein activates c-Met signaling [214,215,216]. Immunohistochemically, ASPS shows nuclear immunoreactivity for TFE3 [217,218]. The immunopositivity for cathepsin K (100%) is also typical. In addition, calretinin (46%) [219,220] and focal desmin (50%) are also expressed in ASPS cases [221].

Although the *ASPSCR1*::*TFE3* fusion in sarcomas appears highly specific and sensitive for ASPS, the same gene fusion is also found in a small subset of *TFE3*-rearranged renal cell carcinomas that affect young patients and have a morphology similar to ASPS [222]. In a clinical study with the c-Met inhibitor crizotinib in ASPS, disease stabilization was reported in most *TFE3*-rearranged ASPS *MET*-altered patients [223]. Therefore, c-MET could be a potential therapeutic target in ASPS.

### 13.5. Desmoplastic Small Round Cell Tumor

Desmoplastic small round cell tumor (DSRCT) is a malignant mesenchymal neoplasm of primitive small round tumor cells associated with prominent desmoplastic stroma and polyphenotypic differentiation [224]. It is characterized by a recurrent chromosomal translocation t(11;22)(p13;q12), resulting in the fusion of the *EWSR1* gene on 22q12.2 and the *WT1* gene on 11p13 [225,226,227,228]. The aberrant transcription of *EWSR1*::*WT1* regulates the expression of various genes and activates the neural reprogramming factor ASCL1 to induce partial neural differentiation [229,230]. Immunohistochemically, DSRCT shows a characteristic polyphenotypic profile expressing epithelial, muscular, and neural markers. A polyclonal antibody to the carboxy (C)-terminus of WT1 is reactive and useful for diagnosis [231].

Recently, there was a report of three unusual tumors affecting the female genital tract with *EWSR1*::*WT1* gene fusion lacking features of DSRCT [232]. These findings suggest the pleiotropy of the *EWSR1*::*WT1* fusion is possible and not restricted to DSRCT. Detection of the *EWSR1*::*WT1* gene fusion can be particularly useful in cases with unusual clinical or histological features [233]. DSRCT is an aggressive disease, despite multimodal therapies [234]. Hence, a better understanding of disease biology is necessary for identifying potential targets in the future [235].

### 13.6. Intimal Sarcoma

Intimal sarcomas are malignant mesenchymal tumors arising within the large blood vessels of the systemic and pulmonary circulations and in the heart [236]. The defining features are primarily intraluminal growth, obstruction of the lumen in the originating vessel, and seeding of tumor emboli in peripheral organs. Genetically, frequent amplifications/gains in the 12q13–q14 region (which contains *MDM2* and *CDK4*) and (co)amplification/gains of *PDGFRA, EGFR,* and *KIT* are present [237,238,239,240]. MDM2 and PDGFR pathways may play a role in the pathogenesis of intimal sarcoma. Immunohistochemically, nuclear expression of MDM2 is observed in at least 70% of cases [237,240]. In addition, rare cases containing rhabdomyosarcomatous differentiation are positive for myogenin and MyoD1 [42].

Intimal sarcomas and undifferentiated cardiac sarcomas carry mutually exclusive *MDM2, MDM4,* and *CDK6* amplifications and share a typical DNA methylation signature [241]. Many primary cardiac sarcomas with histological features of undifferentiated pleomorphic sarcoma are currently reported as intimal sarcomas, especially if there is *MDM2* expression [242].

Table 1 is a summary of the recently described molecular genetic changes and related immunohistochemical markers in some soft tissue tumors.

## 14. Undifferentiated Small Round Cell Sarcomas

### 14.1. Ewing Sarcoma

Ewing sarcoma (EWS) is a small round cell sarcoma characterized by a fusion of a *FET* gene family member (most commonly *EWSR1*) and an *ETS* gene family member [243]. Further mutations can occur in *STAG2* (15–22%), *CDKN2A* (12%), and *TP53* (7%) [244,245,246]. *FET*::*ETS* fusion genes encode chimeric transcription factors that function as master regulators to activate and repress thousands of genes. Expression of these aberrant transcription factors is required to develop EWS. The most common translocation (in 85–90% of cases) is t(11;22)(q24;q12), which results in the *EWSR1*::*FLI1* fusion transcript and protein. The second most common is t(21;22)(q22;q12), which results in *EWSR1::ERG* in 5–10% of EWS cases. Immunohistochemically, strong, diffuse membranous expression of CD99 is observed in approximately 95% of EWSs. NKX2-2, a neuroendocrine/glial transcription factor, has a higher specificity than CD99 [247] (Figure 8). Strong nuclear ERG immunoreactivity is observed in EWS cases with *EWSR1*::*ERG* rearrangement [248]. FLI1 is expressed in the majority of EWS cases, regardless of the fusion variant. The adamantinoma-like variant of EWS consistently demonstrates diffuse cytokeratin, p63, and p40 positivity [249].

Because *EWSR1* fusions are identified in a diverse array of tumor types, FISH for *EWSR1* is not specific for EWS; nonetheless, in the appropriate context, demonstration of *EWSR1* rearrangement is sufficient to confirm the diagnosis [250]. A FISH-based approach using break-apart probes for *EWSR1* and/or *FUS* uncovers most EWS cases. However, a minority of cases with complex inversion/insertion, structural rearrangements, or cryptic insertions may be negative by FISH and require RNA-based techniques for molecular diagnosis [251,252].

### 14.2. Round Cell Sarcoma with EWSR1-Non-ETS Fusions

Round cell sarcomas with *EWSR1*–non-ETS fusions are round, and spindle cell sarcomas with *EWSR1* or *FUS* fusions involving partners irrelevant to the *ETS* gene family [253]. This category includes *EWSR1/FUS*::*NFATC2* sarcomas [254] and *EWSR1*::*PATZ1* sarcomas [255]. *EWSR1/FUS*::*NFATC2* sarcomas have a preference for the bone and consist of round to spindle cells arranged in cords, nests, and trabeculae on a myxohyaline background. *EWSR1*::*PATZ1* sarcomas usually occur in the deep soft tissue and have diverse morphologic features, with a small round to spindle cells in the fibrous stroma, variable necrosis, and mitoses. Immunohistochemically, CD99 is diffusely expressed in 50% of *EWSR1/FUS*::*NFATC2* sarcoma cases. NKX2-2 and PAX7 may also be expressed [256,257]. AGGRECAN shows diffuse, cytoplasmic, and membranous staining [258]. NKX3-1 is frequently expressed in *EWSR1*::*NFATC2* sarcomas [259,260]. *EWSR1*::*PATZ1* sarcomas exhibit co-expression of myogenic markers (desmin, myogenin, MyoD1) and neural markers (S100 protein, SOX10, GFAP) [255].

The *EWSR1/FUS*::*NFATC2* and *EWSR1*::*PATZ1* fusions can be identified via diverse molecular approaches [261,262]. However, *EWSR1*::*PATZ1* fusion is easily missed when using *EWSR1* break-apart FISH [263]. Currently, NGS-based fusion panels are often applied to identify these gene rearrangements and confirm the diagnosis [250].

### 14.3. CIC-Rearranged Sarcoma

*CIC*-rearranged sarcoma is a high-grade undifferentiated round cell sarcoma defined by *CIC*-related gene fusions, most often *CIC*::*DUX4* [264]. A *CIC*::*DUX4* fusion is present in 95% of cases, resulting from either a t(4;19)(q35;q13) or a t(10;19)(q26;q13) translocation [265,266]. Rare cases are associated with non-*DUX4* partner genes, including *FOXO4, LEUTX, NUTM1,* and *NUTM2A* [267,268,269]. The *CIC*::*DUX4* fusion significantly enhances the CIC transcriptional activity and upregulates its targets, including *CCND2, MUC5AC*, and PEA3 family genes (e.g., *ETV1, ETV4,* and *ETV5*) [265,270]. In addition, the *CIC*::*DUX* sarcomas demonstrate frequent *MYC* amplification [271]. Immunohistochemically, DUX4 is a highly sensitive and specific marker for the differentiation of sarcoma with *CIC*::*DUX4* fusion from its histologic mimics [272]. WT1 (anti-N-terminus monoclonal antibodies) (90–95%) and ETV4 (95–100%) are frequently positive and represent useful ancillary markers [273,274]. NKX2-2 is negative in *CIC*-rearranged sarcomas [275]. Sarcomas with *CIC*::*NUTM1* fusions express NUT protein [267].

*CIC*-mutated or rearranged angiosarcomas represent a potential diagnostic pitfall [269]. *CIC* fusions have also been described in central nervous system tumors [276]. *CIC*-rearranged sarcomas are aggressive tumors with frequent metastases and poor outcomes. Their five-year survival rate ranges from 17–43%, and the response to standard EWS chemotherapy regimens is generally poor [267,277].

### 14.4. Sarcoma with BCOR Genetic Alteration

Sarcomas with *BCOR* genetic alterations are clinically distinct sarcomas arising in soft tissue and bone and divided into two main groups. The first group is characterized by sarcomas with *BCOR*-related gene fusions (*BCOR*-fusion sarcomas), most frequently *BCOR*::*CCNB3*. The second group shows internal tandem duplication in *BCOR* (*BCOR*-ITD sarcomas), described in infantile undifferentiated round cell sarcomas and primitive myxoid mesenchymal tumors of infancy [278,279]. *BCOR*-fusion and *BCOR*-ITD sarcomas show a similar gene expression signature [280,281,282,283]. Immunohistochemically, all tumors with various *BCOR* gene alterations show strong and diffuse nuclear positivity for BCOR (Figure 9). BCOR immunostaining may be diagnostically useful but is not specific [282,284]. Additionally, SATB2, TLE1, and cyclin D1 expressions are present in most sarcomas with *BCOR* genetic alterations. *BCOR*::*CCNB3* sarcomas also express CCNB3 [284,285]. Pediatric soft tissue tumors with *BCOR*-ITD show membranous and cytoplasmic expression of EGFR [286].

*BCOR* family tumors share a morphologic spectrum with a similar immunoprofile and gene expression, suggesting a shared pathogenesis. Since immunohistochemistry for either BCOR or CCNB3 is not completely sensitive and specific, a molecular genetic approach is necessary for diagnosis [284]. *BCOR*::*CCNB3* sarcomas often respond to EWS regimens and have a similar outcome [283]. The outcomes of the other *BCOR* family tumors need to be better defined.

Table 2 shows a summary of molecular genetic alterations and immunohistochemical markers in undifferentiated small round cell sarcomas.

## 15. Emerging Entities

### 15.1. EWSR1::SMAD3–Positive Fibroblastic Tumor

*EWSR1*::*SMAD3*–positive fibroblastic tumor (ESFT) is a benign neoplasm defined by a fusion of exon 7 of *EWSR1* with exon 5 of *SMAD3* [287,288,289]. It is characterized by small dermal and subcutaneous acral nodules and histological zonation with an acellular hyalinized center and peripheral fascicular spindle cell growth [290]. SMAD3 is a critical signal transducer in the TGF-β/SMAD signaling pathway involved in extracellular matrix synthesis by fibroblasts. Immunohistochemically, the fibroblastic tumor cells consistently show diffuse ERG nuclear expression, which correlates with a significant *ERG* mRNA upregulation [289].

At the transcriptional level, ESFTs also show overexpression of *FN1* (fibronectin) [289]. Based on molecular data and morphologic and clinical similarities, ESFTs have been suggested to be associated with calcifying aponeurotic fibroma, lipofibromatosis, and lipofibromatosis-like neural tumors [289,290]. The diagnosis is primarily based on the detection of *EWSR1*::*SMAD3* fusion [291]. The terminology of this tumor is provisional while the specific features await further determination.

### 15.2. NTRK-Rearranged Spindle Cell Neoplasm

NTRK-rearranged spindle cell neoplasms (other than infantile fibrosarcomas) are an emerging family of rare spindle cell neoplasms defined by *NTRK* fusions [292]. The tumors are characterized by haphazardly arranged monomorphic spindle cells, infiltrative growth in adipose tissue resembling lipofibromatosis, and characteristic stromal and perivascular keloid collagen. Most tumors harbor *NTRK1* fusions with various partners, including *LMNA, TPR,* or *TPM3* [293,294,295]. Rare cases with *NTRK2* and *NTRK3* fusions have also been reported [296]. The NTRK fusions lead to the activation of the oncogenic signaling pathway via chimeric proteins containing the tropomyosin receptor kinase domains of TRK-A, TRK-B, and TRK-C [297]. Immunohistochemically, most tumors with NTRK fusions are reactive with a monoclonal anti–pan-TRK antibody [117]. The staining can be either cytoplasmic or nuclear [115,298]. Most tumors show frequent coexpression of S100 protein and CD34. TRK-A immunohistochemistry is useful for the detection of *NTRK1*-rearranged tumors [299].

NTRK fusions have been reported at high frequency in various cancers, such as IFS and secretory breast carcinoma, and at a low frequency in other well-established tumors, such as GISTs [300,301]. Importantly, pan-TRK and TRK-A immunoreactivity are not completely specific, and additional molecular genetic testing is often required for a conclusive diagnosis. Hence, molecular detection of NTRK fusions can be useful in such cases [294,302]. Recently, NTRK-rearranged spindle cell neoplasms are ubiquitous tumors of myofibroblastic lineage with a distinct methylation class [303]. Our understanding of this tumor type is rapidly evolving. Additional genetic alterations will be discovered in further studies.

### 15.3. SWI/SNF Complex-Deficient Neoplasms

The SWItch/sucrose non-fermentable (SWI/SNF) complexes are a family of multi-subunit complexes that use the energy of adenosine triphosphate hydrolysis to remodel nucleosomes [304]. Chromatin remodeling processes mediated by the SWI/SNF complexes are critical to the modulation of gene expression across various cellular processes, including stemness, differentiation, and proliferation. Recently, mutations in the genes encoding different subunits of the SWI/SNF complex (*SMARCB1, SMARCA4, ARID1A, ARID1B, ARID1,* and *PBRM*) have been identified in many adult malignancies, which include encompassing epithelial and mesenchymal tumors [305,306]. Several SMARCA4-deficient poorly differentiated and undifferentiated carcinomas/sarcomas delineated primarily based on BRG1 protein loss have been characterized in various anatomical locations [307].

Thoracic SMARCA4-deficient undifferentiated tumor is a high-grade malignant neoplasm significantly affecting the thorax in adults. It shows an undifferentiated or rhabdoid phenotype and a lack of SMARCA4 (BRG1), a vital member of the SWI/SNF chromatin-remodeling complex [308]. The tumor is driven by biallelic inactivation of *SMARCA4,* including mainly nonsense and frameshift mutations. Histologically, the tumor consists of diffuse sheets of monomorphic, undifferentiated epithelioid cells with vesicular nuclei and prominent nucleoli, and frequent rhabdoid features. Immunohistochemically, complete loss of SMARCA4 (BRG1) expression is typical (Figure 10). Additionally, there is a family of tumors defined by SMARCA4 loss, including small cell carcinoma of the ovary, hypercalcemic type [309], and large cell malignancies originating in the uterus [310]. Recently, SMARCA4-deficient sinonasal carcinomas have also been reported [311].

### 15.4. DICER1-Associated Sarcomas

The *DICER1* gene is located on chromosome 14q32.13 and encodes an endoribonuclease in the ribonuclease (RNase) III family required for processing microRNA (miRNA) [312]. Dysregulation of microRNA by *DICER1* mutations causes activation of oncogenes and underlies developmental and neoplastic disorders. A wide variety of *DICER1*-associated sarcomas have been reported in the literature [313]. *DECER1*-associated sarcomas, regardless of their site of origin, exhibit characteristic morphology that resembles pleuropulmonary blastoma [314]. The morphologic features include a subepithelial layer of malignant mesenchymal cells (cambium layer), areas of rhabdomyoblastic differentiation with positive staining for myogenin and myoD1, cellular/immature and occasionally malignant cartilage, bone/osteoid foci, and areas of anaplasia [315]. If a mesenchymal neoplasm is reported that has a combination of rhabdomyoblastic, cartilaginous, and neuroectodermal elements, a *DICER1*-associated neoplasm should be considered.

Kommoss et al. [316] reported the clinicopathological and molecular features of *DICER1*-mutant and *DICER1*-wild-type embryonal RMS in a series of genitourinary tumors. They suggested that *DICER1*-mutant ERMS might represent a distinct subtype in the future classification of RMS. Unsupervised hierarchical clustering of array-based whole-genome methylation data of a subset of *DICER1*-mutant sarcomas has revealed that they cluster together [317]. Further methylation studies are warranted to determine whether *DICER1*-mutant sarcomas constitute a distinct, identifiable subclass of sarcomas.

Table 3 shows a summary of molecular genetic alterations and immunohistochemical markers in emerging entities.

## 16. Conclusions

Advances in molecular techniques have refined the classification of soft tissue tumors. This review summarizes newly recognized and emerging entities, focusing on molecular alterations and antibodies as surrogate markers for molecular genetic techniques. In addition, already well-defined entities with recently described molecular changes are discussed. The critical genetic events driving the biology of soft tissue tumors are still largely unknown. Further studies with careful genomic-morphologic correlation are required to understand soft tissue pathology comprehensively.

## Figures and Tables

**Figure 1 ijms-24-05934-f001:**
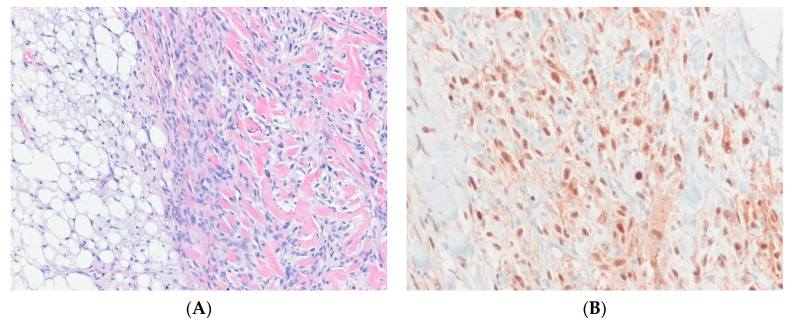
Dedifferentiated liposarcoma. (**A**) The tumor shows an abrupt transition from a well-differentiated liposarcoma component (left) to a non-lipogenic sarcoma component (right), MDM2 (H&E stain, ×100). (**B**) The dedifferentiated part shows nuclear positivity for MDM2, indicating the presence of underlying *MDM2* amplification (immunohistochemical stain for MDM2, ×200).

**Figure 2 ijms-24-05934-f002:**
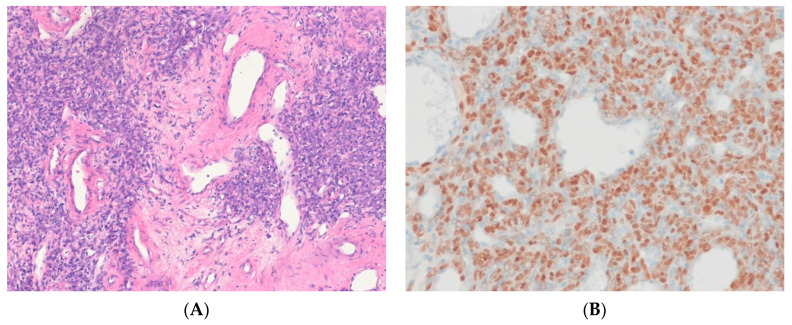
Solitary fibrous tumor. (**A**) The tumor shows spindle to ovoid cells with hypocellular and hypercellular areas and a hemangiopericytoma-like vascular pattern (H&E stain, ×100). (**B**) The tumor cells show diffuse nuclear positivity for STAT6, suggesting the presence of an underlying *NAB2*::*STAT6* fusion gene (immunohistochemical stain for STAT6, ×200).

**Figure 3 ijms-24-05934-f003:**
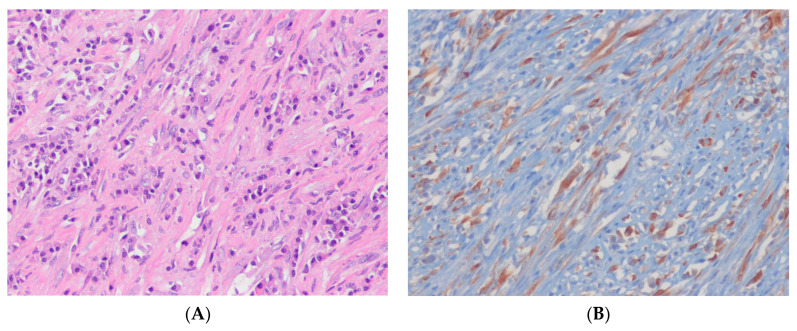
Inflammatory myofibroblastic tumor. (**A**) The tumor shows spindle cells with vesicular nuclei and eosinophilic cytoplasm admixed with lymphocytes and plasma cells (H&E stain, ×200). (**B**) The tumor cells show cytoplasmic positivity for ALK*,* indicating the presence of an underlying *ALK* rearrangement (immunohistochemical stain for ALK, ×200).

**Figure 4 ijms-24-05934-f004:**
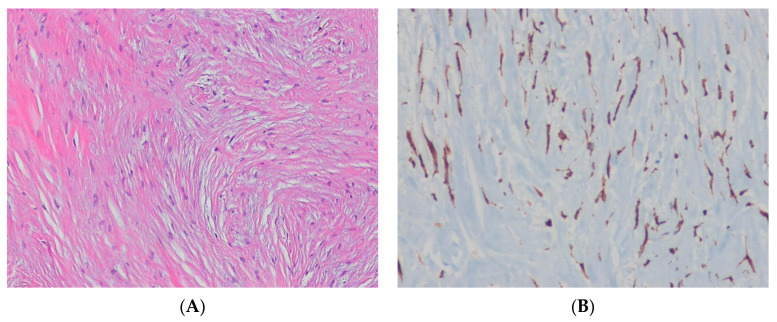
Low-grade fibromyxoid sarcoma. (**A**) The tumor shows bland-appearing spindle cells in the myxoid and fibrous stroma, with short fascicles or a storiform pattern (H&E stain, ×100). (**B**) The tumor cells show cytoplasmic expression for MUC4, indicating the presence of underlying *MUC4* upregulation (immunohistochemical stain for MUC4, ×200).

**Figure 5 ijms-24-05934-f005:**
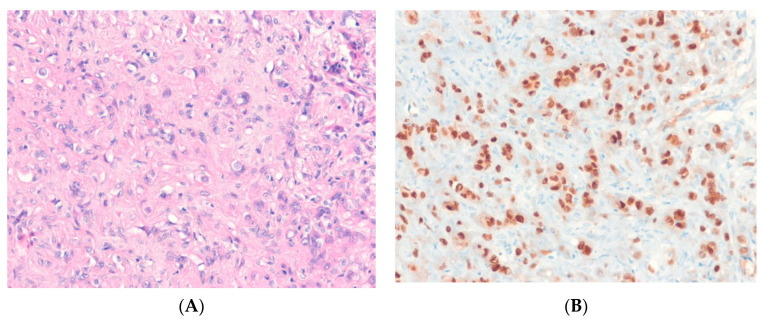
Epithelioid hemangioendothelioma. (**A**) The tumor cells show epithelioid cells with oval or round nuclei and variable amounts of eosinophilic cytoplasm in myxohyaline stroma. Intracytoplasmic vacuoles are present (H&E stain, ×100). (**B**) The tumor cells show strong nuclear expression of CAMTA1, indicating the presence of an underlying *WWTR1*::*CAMTA1* fusion gene (immunohistochemical stain for CAMTA1, ×200).

**Figure 6 ijms-24-05934-f006:**
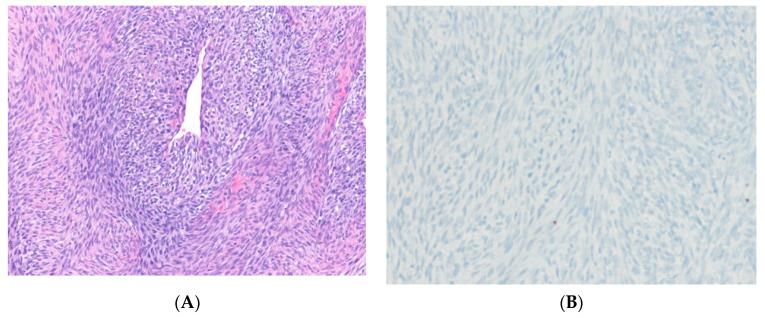
Malignant peripheral nerve sheath tumor. (**A**) The tumor shows spindle cells in a fascicular pattern. Perivascular accentuation of cellularity is present (H&E stain, ×100). (**B**) The tumor cells show loss of H3K27me3 expression, indicating the presence of underlying inactivating mutations of the polycomb repressive complex 2 components (*SUZ12* or *EED*) (immunohistochemical stain for H3K27me3, ×200).

**Figure 7 ijms-24-05934-f007:**
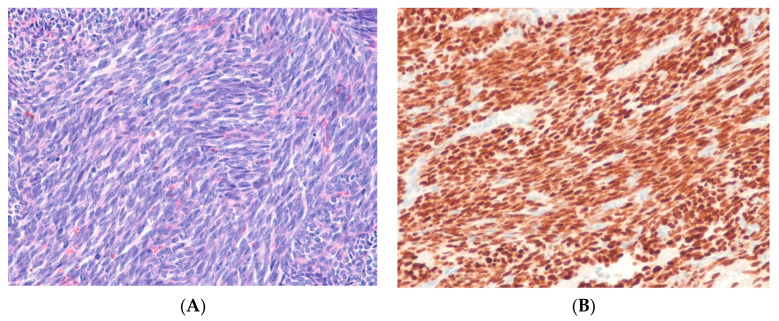
Monophasic synovial sarcoma. (**A**) The tumor shows fascicles of monomorphic spindle cells (H&E stain, ×100). (**B**) The tumor cells show strong nuclear expression for SS18-SSX antibody, consistent with the presence of an underlying *SS18*::*SSX* fusion gene (immunohistochemical stain for SS18-SSX antibody, ×200).

**Figure 8 ijms-24-05934-f008:**
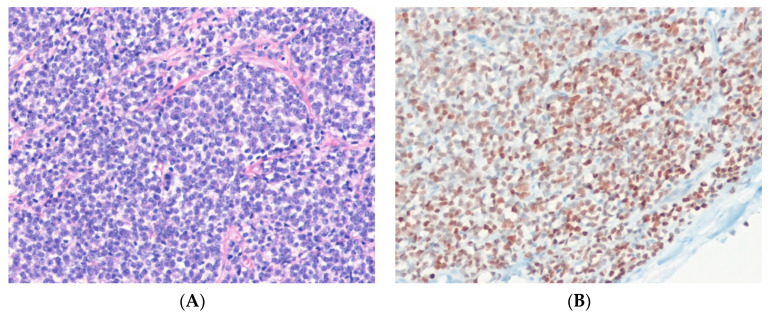
Ewing sarcoma. (**A**) The tumor shows round cells with finely granular chromatin and clear cytoplasm (H&E stain, ×100). (**B**) The tumor cells show diffuse nuclear staining for NKX2-2, suggesting the presence of underlying *NKX2*-*2* upregulation as a downstream target of the *EWSR1*::*FLI1* fusion gene (immunohistochemical stain for NKX2-2, ×100).

**Figure 9 ijms-24-05934-f009:**
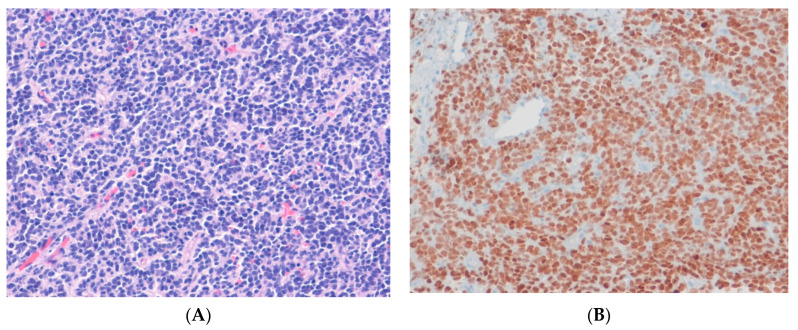
*BCOR*-rearranged sarcoma. (**A**) The tumor shows solid sheets of monomorphic round cells with round nuclei and scant eosinophilic cytoplasm, with delicate capillaries (H&E stain, ×100). (**B**) The tumor cells show diffuse nuclear staining for BCOR, indicating the presence of an underlying *BCOR*::*CCNB3* fusion gene (immunohistochemical stain for BCOR, ×200).

**Figure 10 ijms-24-05934-f010:**
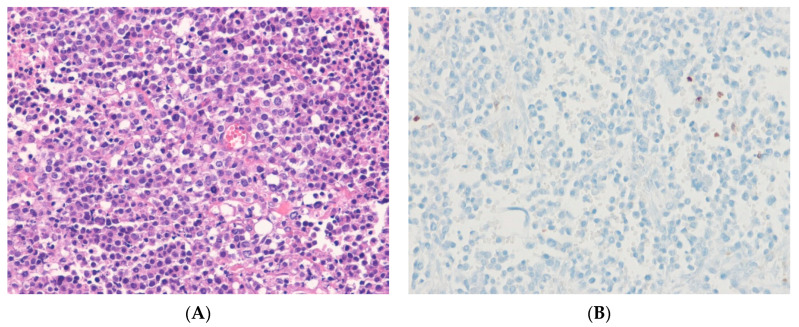
Thoracic SMARCA4-deficient undifferentiated tumor. (**A**) The tumor shows diffuse sheets of epithelioid tumor cells with uniform round nuclei and prominent nucleoli (H&E stain, ×200). (**B**) Most tumor cells show complete loss of SMARCA4, indicating the presence of underlying *SMARCA4* inactivation (immunohistochemical stain for SMARCA4, ×200).

**Table 1 ijms-24-05934-t001:** Recently described molecular genetic alterations and immunohistochemical markers in selected soft tissue tumors.

Tumor Category	Tumor Type	Cytogenetic Alterations	Molecular Alterations	Immunohistochemical Markers	Staining Pattern	References
Adipocytic tumors	Spindle cell/pleomorphic lipoma	Loss of 13q14 Loss of 16q	*RB1* deletion	RB1	Loss	[20,21,22,23]
	Atypical spindle cell/pleomorphic lipomatous tumor	Loss of 13q14 Monosomy 7	*RB1* deletion (subset)	RB1	Loss	[28,29,30,31]
	Atypical lipomatous tumor/well-differentiated liposarcoma	Gain of 12q13–15 region	*MDM*2 amplification *CDK4* amplification	MDM2, CDK4	Nuclear staining	[35,36]
	Dedifferentiated liposarcoma	Gain of 12q13–15 region	*MDM*2 amplification *CDK4* amplification	MDM2, CDK4	Nuclear staining	[45,46]
	Myxoid liposarcoma	t(12;16)(q13;p11) t(12;22)(q13;q12)	*FUS*::*DDIT3* (>90%) *EWSR1*::*DDIT3* (3%)	DDIT3	Nuclear staining	[50,56]
Fibroblastic and myofibroblastic tumors	Desmoid fibromatosis	Trisomy 8, trisomy 20 Loss of 5q21	*CTNNB1* mutation *APC* mutation	β-catenin	Nuclear staining	[60,61,65,66]
	Solitary fibrous tumor	inv12(q13;q13)	*NAB2*::*STAT6*	STAT6	Nuclear staining	[72,73,74,80]
	Inflammatory myofibroblastic tumor	t(1;2)(q21;p23) t(2;19)(p23;p13) t(2;17)(p23;q23) t(6;17)(q22;p13) t(3;6)(q12;q22)	*TPM3*::*ALK* *TPM4*::*ALK* *CLTC*::*ALK* *ROS*1::*YWHAE* *ROS1*::*TFG1*	ALK ROS1	Cytoplasmic staining	[85,86,87,88,89]
	Epithelioid inflammatory myofibroblastic sarcoma	t(2;2)(p23;q13)	*RANBP2*::*ALK* *RRBP1*::*ALK*	ALK	Nuclear membrane or perinuclear accentuation	[90,91]
	Low-grade fibromyxoid sarcoma	t(7:16)(q33;p11) t(11;16)(p11;p11)	*FUS*::*CREB3L2* (>90%) *FUS*::*CREB3L1* *EWSR1*::*CREB3L1*	MUC4	Cytoplasmic staining	[94,95,96,97,98,99,100]
	Sclerosing epithelioid fibrosarcoma	t(11;22)(p11;q12) t(11;16)(p11;p11) t(7;16)(q34;p11)	*EWSR1*::*CREB3L1* (80–90%) *EWSR1*::*CREB3L2* *FUS*::*CREB3L2*	MUC4	Cytoplasmic staining	[102,103,104,105,106,107]
	Infantile fibrosarcoma	t(12;15)(p13;q25)	*ETV6*::*NTRK3* *EML4*::*NTRK3*	Pan-TRK	Cytoplasmic or membranous staining	[112,113,114,115]
Vascular tumors	Epithelioid hemangioma	t(19;19)(q13;q13) t(7;19)(q22;q13)	*FOS*::*VIM, FOS*::*LMNA*, *ZFP36*::*FOSB, WWTR1*::*FOSB*	FOS (subset) FOSB (subset)	Nuclear staining	[120,121,122,123]
	Pseudomyogenic hemangioendothelioma	t(7;19)(q22;q13)	*SERPINE1*::*FOSB* *ACTB*::*FOSB*	FOSB	Nuclear staining	[126,127,128]
	Epithelioid hemangioendothelioma	t(1;3)(p36;q25) t(X;11)(p11;q13)	*WWTR1*::*CAMTA1* (85–90%) *YAP1*::*TFE3* (5%)	CAMTA1 TFE3	Nuclear staining	[134,135,136,137,138,139,140]
Skeletal muscle tumors	Alveolar rhabdomyosarcoma	t(2;13)(q35;q14) t(1;13)(p36;q14)	*PAX3*::*FOXO1* (70–90%) *PAX7*::*FOXO1* (10–30%)	Myogenin MyoD1	Nuclear staining	[142,143,148,149]
	Congenital/infantile spindle cell rhabdomyosarcoma		*SRF*::*NCOA2, TEAD1*::*NCOA2, VGLL2*::*NCOA2, VGLL2*::*CITED2*	Myogenin MyoD1	Nuclear staining	[153,154]
	*MYOD1*-mutant spindle cell/sclerosing rhabdomyosarcoma		*MYOD1* mutations	Myogenin MyoD1	Nuclear staining	[154,155]
	Intraosseous spindle cell rhabdomyosarcoma		*EWSR1/FUS::TFCP2* *MEIS1*::*NCOA2*	Myogenin MyoD1	Nuclear staining	[155,156]
Gastrointestinal stromal tumors	Gastrointestinal stromal tumor		*KIT* mutations (75%) *PDGFRA* mutations (10%)	KIT (CD117) DOG1	Cytoplasmic or membranous staining	[160,161,162,163,169]
	SDH-deficient gastrointestinal stromal tumor		*SDH* mutations	SDHB SDHA	Loss	[165,166,168]
Peripheral nerve sheath tumor	MPNST	Complex karyotype with numerical and structural abnormalities	*NF1* inactivation; PRC2 components (*EED* or *SUZ12)* inactivation	H3K27me3	Loss	[177,178,179]
	Epithelioid MPNST	22q deletion	*SMARCB1* inactivation	SMARCB1	Loss	[180]
	MMNST	Mutation and/or loss of heterozygosity of 17q	*PRKAR1A* inactivation	PRKAR1A	Loss	[183,184]
Tumors of uncertain differentiation	Synovial sarcoma	t(X;18)(p11.2;q11.2)	*SS18*::*SS1, SS2,* or *SS4* fusion; *SS18L1*::*SSX1* fusion (rare)	SS18-SSX	Nuclear staining	[189,190,193,194]
	Epithelioid sarcoma	22q11.2 deletion	*SMARCB1* inactivation	SMARCB1 (INI1)	Loss	[197,199,200,201]
	Extrarenal rhabdoid tumor	22q11.2 deletion	*SMARCB1* inactivation	SMARCB1 (INI1)	Loss	[206,207]
	Alveolar soft part sarcoma	der(17)t(X;17)(p11;q25)	*ASPSCR1*::*TFE3*	TFE3	Nuclear staining	[217,218]
	DSRCT	t(11;22)(p13;q12)	*EWSR1*::*WT1*	WT1 (C-erminus)	Nuclear staining	[225,226,227,228,231]
	Intimal sarcoma	12q12-15 amplification	*MDM2* amplification *PDGFRA* amplification	MDM2	Nuclear staining	[237,238,239,240]

MPNST, malignant peripheral nerve sheath tumor; PRC2, polycomb repressive complex 2; MMNST, malignant melanotic nerve sheath tumor; DSRCT, desmoplastic small round cell tumor; SDH, succinate dehydrogenase.

**Table 2 ijms-24-05934-t002:** Molecular genetic alterations and immunohistochemical markers in undifferentiated small round cell sarcomas.

Tumor Type	Cytogenetic Alterations	Molecular Alterations	Immunohistochemical Markers	Staining Pattern	References
Ewing sarcoma	t(11;22)(q24;q12) t(21;22)(q22;q12) t(2;22)(q33;q12) t(7;22)(p22;q12) t(17;22)(q21;q12)	*EWSR1*::*FLI1* (85–90%) *EWSR1*::*ERG* (5–10%) *EWSR1*::*FEV* *EWSR1*::*ETV1* *EWSR1*::*ETV4*	NKX2-2 FLI1 ERG CD99	NKX2-2, FLI1, ERG, nuclear staining; CD99, membranous staining	[244,245,246,247,248]
Round cell sarcoma with *EWSR1*–non-ETS fusions	*EWSR1/FUS*::*FATC2* sarcoma; t(20;22)(q13;q12) t(16;20)(p11;q13) *EWSR1*::*PATZ1* sarcoma; t(22;22)(q12;q12)	*EWSR1/FUS*::*FATC2* sarcoma; *EWSR1*::*NFATC2* *FUS*::*NFATC2* *EWSR1*::*PATZ1* sarcoma; *EWSR1*::*PATZ1*	*EWSR1/FUS*::*FATC2* sarcoma; AGGRECAN NKX3-1 *EWSR1*::*PATZ1* sarcoma; Myogenic (myogenin, MyoD1) and neurogenic markers (S100 protein)	AGGRECAN, cytoplasmic staining; NKX3-1, myogenin, myoD1, nuclear staining; S100 protein, nuclear and cytoplasmic staining	[230,255,259]
*CIC*-rearranged sarcoma	t(4;19)(q35;q13) t(10;19)(q26;q13) t(X;19)(q13;q13) t(15;19)(q14;q13)	*CIC*::*DUX4* (95%) *CIC*::*FOXO4* *CIC*::*LEUTX* *CIC*::*NUTM1* *CIC*::*NUTM2A*	DUX4 ETV4 WT1 (N-terminus) NUT	Nuclear staining	[265,266,267,268,269,272,273,274]
Sarcoma with *BCOR* genetic alterations	inv(X)(p11.4p11.22) t(X;22)(p11q13) t(10;17)(q22;p13)	*BCOR*::*CCNB3, BCOR*::*MAML3*, *BCOR*::*ZC3H7B*; *BCOR* internal tandem duplications	BCOR CCNB3 SATB2	Nuclear staining	[282,283,284,285]

**Table 3 ijms-24-05934-t003:** Molecular genetic alterations and immunohistochemical markers in emerging entities.

Tumor Type	Cytogenetic Alterations	Molecular Alterations	Immunohistochemical Markers	Staining Pattern	References
*EWSR1*::*SMAD3*–positive fibroblastic tumor	t(15;22)(q22.33;q12.2)	*EWSR1*::*SMAD3* fusion	ERG	Nuclear staining	[289]
NTRK-rearranged spindle cell neoplasm		*NTRK1* fusions with *LMNA, TPR,* or *TPM3*; *NTRK2, NTRK3* fusions	Pan-TRK TRK-A	Cytoplasmic or nuclear staining	[115,117,293,294,295,296,298],
Thoracic SMARCA4-deficient undifferentiated tumor		Biallelic inactivation of *SMARCA4*	SMARCA4 (BRG1)	Loss	[307]
*DICER1*-associated sarcoma		*DICER1* mutations	Myogenin MyoD1	Nuclear staining	[315]

## Data Availability

Not applicable.

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
