# Peer review of "The Recent Advances in Molecular Diagnosis of Soft Tissue Tumors"

_ijms, 2023, doi:10.3390/ijms24065934_

Round 1
Reviewer 1 Report
Choi and Ro summarized recent progress on diagnostic markers for soft tissue tumors with the aids from new molecular and genetic tools. Tumor diagnosis in general is very important and challenging due to its complexity and lack of accurate tools, this manuscript provided comprehensive and extensive information among major and subtypes of soft tissue tumors, which could potentially guide both clinicians and scientists with methods, markers and current knowledge of the specific type of interest. In my opinion, there several points need to be addressed and improved to serve better for its purpose, listed as follows.
1. Although the authors implied that information included in the manuscript is ‘recent advance’, there are no clear clues to distinguish ‘traditional/conventional’ from ‘recent’ progress, which applies to sections regarding methods and detailed information of each tumor type. For example, genetic classification utilizes chromosomal translocation, results of which are categorized as 'cytogenetic alterations’ in the tables. But whether or not this can be obtained with conventional methods and new techniques can provide additional/better results are not mentioned and discussed. Next generation sequencing has been mentioned multiple times in the manuscript, however, the advances this method brought to the field and each tumor type are not clearly stated. Additionally, regarding molecular alterations, how FISH, RT-PCR and next generation have been applied and generated insights of each tumor type is not clear.
2. The three tables of the manuscript provide clear summary of major diagnostic markers of the three categories, which is very helpful to the readers to locate the most relevant information. It would be even better if references can be added to the table, so that readers can more easily access supporting evidence.
3. Figures can be improved with better labeling of the images (e.g. gene marker used) and more organized legends.
Author Response
Authors’ response to reviewers
Dear Editor-in-Chief,
I am pleased to resubmit for publication the revised version of ijms-2259682, entitled "Recent Advances in Molecular Genetics and Immunohistochemistry in the Diagnosis of Soft Tissue Tumors". I appreciate the constructive criticism of the reviewers and the editor. The manuscript has been revised in accordance with the reviewers’ suggestions and the editor’s comments. We made major and minor corrections in blue letters to the original manuscript. Our point-to-point responses to reviewers’ recommendations and editor’s comments are below each.
Reviewer 1:
Choi and Ro summarized recent progress on diagnostic markers for soft tissue tumors with the aids from new molecular and genetic tools. Tumor diagnosis in general is very important and challenging due to its complexity and lack of accurate tools, this manuscript provided comprehensive and extensive information among major and subtypes of soft tissue tumors, which could potentially guide both clinicians and scientists with methods, markers and current knowledge of the specific type of interest. In my opinion, there several points need to be addressed and improved to serve better for its purpose, listed as follows.
1) Although the authors implied that information included in the manuscript is ‘recent advance’, there are no clear clues to distinguish ‘traditional/conventional’ from ‘recent’ progress, which applies to sections regarding methods and detailed information of each tumor type. For example, genetic classification utilizes chromosomal translocation, results of which are categorized as 'cytogenetic alterations’ in the tables. But whether or not this can be obtained with conventional methods and new techniques can provide additional/better results are not mentioned and discussed. Next generation sequencing has been mentioned multiple times in the manuscript, however, the advances this method brought to the field and each tumor type are not clearly stated. Additionally, regarding molecular alterations, how FISH, RT-PCR and next generation have been applied and generated insights of each tumor type is not clear.
Response: Thank you for your suggestion. We have mentioned as follows in “4. Molecular Test” section.
“NGS is a highly sensitive method for detecting genetic alterations and can help to diagnose more precisely and to characterize more detailed genetic alterations. And NGS will provide further insight into the pathogenesis of soft tissue tumors and the basis for the development of targeted therapies.”
2) The three tables of the manuscript provide clear summary of major diagnostic markers of the three categories, which is very helpful to the readers to locate the most relevant information. It would be even better if references can be added to the table, so that readers can more easily access supporting evidence.
Response: References have been added in Tables 1, 2, and 3.
3) Figures can be improved with better labeling of the images (e.g. gene marker used) and more organized legends.
Response: Figures have been improved with better labeling of images and organized legends.
Reviewer 2:
1) The name can be better: the recent advances in molecular diagnosis of soft tissue tumors.
Response: We have revised the name as follows.
“The recent advances in molecular diagnosis of soft tissue tumors”
2) This paper is a good review of the implication of Next Generation IHQ. However.
In the introduction could be useful the reference. https://www.ncbi.nlm.nih.gov/pmc/articles/PMC8167394/pdf/pathol-2021-02-70.pdf3.
Introduction, the authors can mention more about the intrinsic complexity and technological complexity. And the challenge that it means for pathologists to do a good diagnosis of sarcomas.
Response: We have mentioned your suggestion as follows.
They present diagnostic challenges for pathologists due to a large number of tumor types, the rarity of each tumor type, and their considerable morphologic diversity and overlap, intrinsic complexity, and technological complexity [2].
- Sbaraglia, M.; Bellan, E.; Dei Tos, A.P. The 2020 WHO Classification of Soft Tissue
Tumours: news and perspectives. Pathologica. 2021, 113, 70–84.
3) The biggest contribution is table 1. It is an extraordinary resume of the different new molecular diagnoses. It can be placed at the end of the principal text, as a resume of the text. Authors must add the references in a new column.
Response: Thank you for your suggestion. Table 1 has been placed at the end of the principal text. We add references in a new column.
4) In the case of RB1, immunohistochemically markers are better if they mention the loss of nuclear RB1 immunoreactivity. The same is needed for the rest of the immunohistochemical markers.
Response: The staining patterns have been added in Table 1, 2, and 3.
5) In some sections the text is confusing, as example, they mention in the section Next Generation Immunohistochemistry
….The diagnostic markers identified by gene expression profiling…. Probably is better to name the section new generation of molecular diagnosis in sarcomas, because IHQ and PCR or sequencing is not the same.
Response: We have changed “5. Next Generation Immunohistochemistry” into
“5. Immunohistochemistry”.
6) The are some letters in the table in grey and others in black, it is a mistake or is it for some special reason? Please clarify.
Response: We have changed grey letters in the Tables into black letters.
7) At the end of the text, it will be extraordinary if the authors give recommendations for the minimum analyses necessary for a pathologist to suspect a sarcoma and send it to a reference center that has all the equipment for molecular analysis. Since in most of the centers of the world, there is no such infrastructure.
Response: Thank you for your suggestions.
Current European Society for Medical Oncology (ESMO) guidelines suggesting that the morphologic and immunohistochemical analyses should be complemented by molecular pathology have been described in 4. Molecular Tests section. We have added as follows.
“If molecular analysis is not available, it is recommended to send it to a reference with all molecular analysis equipment.”
8) References needs cross references as MPDI format
- Burningham, Z.; Hashibe, M.; Spector, L.; Schiffman, J.D. The Epidemiology of Sarcoma. Sarcoma Res.2012, 2, 14. [Google Scholar] [CrossRef] [PubMed]
- Cormier, J.N.; Pollock, R.E. Soft Tissue Sarcomas. CA Cancer J. Clin.2004, 54, 94–109. [Google Scholar] [CrossRef] [PubMed]
- Bourcier, K.; Le Cesne, A.; Tselikas, L.; Adam, J.; Mir, O.; Honore, C.; de Baere, T. Basic Knowledge in Soft Tissue Sarcoma. Intervent. Radiol.2019, 42, 1255–1261. [Google Scholar] [CrossRef]
- Elkrief, A.; Alcindor, T. Molecular Targets and Novel Therapeutic Avenues in Soft-Tissue Sarcoma. Oncol.2020, 27, 34–40. [Google Scholar] [CrossRef] [PubMed]
Response: We have added DOI of references.
Editor comments:
There are some other issues in your manuscript that require your
attention. Please find them below and revise accordingly during revision:
1) A high repetition rate was detected, please reduce the repetition
rate according to the attachment.
Response: We have reduced the repetition rate.
2) We found that the figures in your manuscript cited some information
from other sources. We wish to confirm that there is no potential
copyright issue with these citations, please kindly let us know whether
it is the case.
Response: All figures had not been previously published. I confirm that there is no copyright issue.
I would appreciate if you could review the manuscript for publication in the International Journal of Molecular Sciences.
Sincerely yours,
Joon Hyuk Choi, MD, PhD
Professor
Department of Pathology, Yeungnam University College of Medicine
170 Hyeonchung-ro, Namgu, Daegu City 42415, Korea, Tel: 82-53-640-6754,
Fax: 82-53-622-8432, E-mail: joonhyukchoi@ynu.ac.kr

Reviewer 2 Report
The name can be better: the recent advances in molecular diagnosis of soft tissue tumors.
This paper is a good review of the implication of Next Generation IHQ. However.
In the introduction could be useful the reference. https://www.ncbi.nlm.nih.gov/pmc/articles/PMC8167394/pdf/pathol-2021-02-70.pdf
Introduction, the authors can mention more about the intrinsic complexity and technological complexity. And the challenge that it means for pathologists to do a good diagnosis of sarcomas.
The biggest contribution is table 1. It is an extraordinary resume of the different new molecular diagnoses. It can be placed at the end of the principal text, as a resume of the text. Authors must add the references in a new column.
In the case of RB1, immunohistochemically markers are better if they mention the loss of nuclear RB1 immunoreactivity. The same is needed for the rest of the immunohistochemical markers.
In some sections the text is confusing, as example, they mention in the section Next Generation Immunohistochemistry
….The diagnostic markers identified by gene expression profiling…. Probably is better to name the section new generation of molecular diagnosis in sarcomas, because IHQ and PCR or sequencing is not the same.
The are some letters in the table in grey and others in black, it is a mistake or is it for some special reason? Please clarify.
At the end of the text, it will be extraordinary if the authors give recommendations for the minimum analyses necessary for a pathologist to suspect a sarcoma and send it to a reference center that has all the equipment for molecular analysis. Since in most of the centers of the world, there is no such infrastructure.
References needs cross references as MPDI format
- Burningham, Z.; Hashibe, M.; Spector, L.; Schiffman, J.D. The Epidemiology of Sarcoma. Clin. Sarcoma Res. 2012, 2, 14. [Google Scholar] [CrossRef] [PubMed]
- Cormier, J.N.; Pollock, R.E. Soft Tissue Sarcomas. CA Cancer J. Clin. 2004, 54, 94–109. [Google Scholar] [CrossRef] [PubMed]
- Bourcier, K.; Le Cesne, A.; Tselikas, L.; Adam, J.; Mir, O.; Honore, C.; de Baere, T. Basic Knowledge in Soft Tissue Sarcoma. Cardiovasc. Intervent. Radiol. 2019, 42, 1255–1261. [Google Scholar] [CrossRef]
- Elkrief, A.; Alcindor, T. Molecular Targets and Novel Therapeutic Avenues in Soft-Tissue Sarcoma. Curr. Oncol. 2020, 27, 34–40. [Google Scholar] [CrossRef] [PubMed]
Author Response

(The authors gave the same response as above.)
